**Subject Category:**
Biology (whole organism)

behaviour/cognition/neuroscience

laboratory mouse strains, inbred, outbred, phenotypic differences, *Disc1* mutation

**Author for correspondence:**
Razia Sultana
e-mail: rrazia1@lsu.edu

# Contrasting characteristic behaviours among common laboratory mouse strains

Razia Sultana[1,2], Olalekan M. Ogundele[2]
and Charles C. Lee[1]

[1]Neural Systems Laboratory, and [2]Synapse Biology Laboratory, Department of Comparative Biomedical Sciences, Louisiana State University School of Veterinary Medicine, Baton Rouge, LA 70803, USA

RS, 0000-0003-3875-2892

Mice are widely used to model wide-ranging human neurological disorders, from development to degenerative pathophysiology. Behavioural and molecular characteristics of these mouse models are influenced by the genetic background of each strain. Among the most commonly used strains, the inbred C57BL/6J, BALB/c, CBA and 129SvEv lines and the CD1 outbred line are particularly predominant. Despite their prevalence, comparative performance of these strains on many standard behavioural tests commonly used to assess neurological conditions remains diffusely and indirectly accessible in the literature. Given that independent studies may be conducted with mice of differing genetic backgrounds, any variation in characteristic behavioural responses of specific strains should be delineated in order to properly interpret results among studies. Thus, in the present study, we aimed to characterize these commonly used mice strains through several standard behavioural tests. Here, we found that animals from different genetic background strains exhibited varying behavioural patterns when assessed for sociability/novelty, memory function, and negative behaviours like despair and stress calls. These results suggest that genetic variation among strains may be responsible—in part—for strain-specific behavioural phenotypes and potential predisposition to some neurological disorders.

## 1. Introduction

Biomedical studies are indebted to animal models as surrogates to illuminate the physiology and pathology of disease. In particular, mouse model systems are heavily employed in neuroscience to study both basic and diseased processes in the nervous system, particularly as they mimic certain basic behavioural phenotypes observed in humans, along with gross and molecular brain

features. Genetically, mice exhibit approximately 90% gene homology with humans [1], with 85% of the protein-coding region of the genome being conserved [2], supporting their validity as models to study the physical, behavioural and molecular changes involved in the pathogenesis of neurodevelopmental and neurodegenerative disorders. Moreover, mice are advantageous due to the constant advent of new techniques and manipulations at genetic and pharmacological levels and due to their small size, high fecundity, easy rearing and management [3]. As such, the mouse model has been instrumental in many aspects of biomedical neuroscience, from understanding disease pathways [4–6] to the development of therapeutic drugs [7,8].

Modelling neurological conditions in mice often involves an assessment of behavioural phenotypes [9,10]. Among the numerous strains available, the C57BL/6J, BALB/c, CBA/J, 129SvEv and CD1 lines are among the most commonly used as controls or as background strains for transgenes [11]. Although there is an overwhelmingly vast literature assessing behaviour in these common mouse lines [9,10,12–14], behavioural comparison among strains remains diffusely and indirectly available [15,16]. This is problematic, since a baseline comparison among strains on common behavioural tests (e.g. sociability, novelty, despair, cognition, anxiety, learning and memory etc.) is necessary to interpret results from different strains [17], particularly as multiple studies conducted under different conditions in separate laboratories could contribute to the variations or similarities among reported strain behaviours [18,19]. Moreover, genetic mutations introduced onto different backgrounds can lead to variable behavioural phenotypes [20] and in part influence conclusions regarding the role of those genes under investigation [21–24].

Indeed, behavioural assays depend, to a degree, upon the genetic background of the experimental animal [25,26], with subtle differences in behaviour known to exist between strains [27–30]. Although such discrepancies among strains may be addressed statistically, such as through consideration of Cohen $d$ effect size [11], using much larger sample size and/or backcrossing transgenic animals on different background strains as control animals, the genetic background still remains a key concern when comparisons are made across studies or with small sample sizes [22–25].

Thus, in this study, we compared the performance of several of the most widely used mouse strains, C57BL/6J, BALB/c, CBA/J and 129SvEv, on several common behavioural test paradigms. Because of the different genetic make-up of outbred lines, when compared with inbred populations, the CD1 outbred strain was also included to provide an additional metric for these behavioural comparisons [31,32]. In addition, 129SvEv strain was chosen with a specific aim in mind, particularly due to its well-described spontaneous mutation in a schizophrenia candidate gene (*Disrupted in Schizophrenia*, aka *Disc1*) and its wide use in a number of studies relevant to its disease mechanisms [33–38].

Our work indicates that baseline behaviours vary among these common laboratory mouse strains, suggesting a framework for interpreting results across studies. Furthermore, our results also highlight background baseline behaviour of strains as a criterion for model selection in biomedical studies involving behaviour and physiology.

# 2. Material and methods

## 2.1. Animal care and housing

A total of 12 animals ($n = 6$ each of males and females) each of C57BL/6J, 129SvEv, BALB/c, CBA and CD-1 mice between 6 and 8 weeks of age obtained from the Jackson laboratory (Bar Harbor, ME), DLAM LSU (Baton Rouge, LA) and Charles River Lab (Wilmington, MA), respectively, were assessed on a battery of behavioural tests. Animals were housed in a temperature- and humidity-controlled room with a 12 h light/dark cycle with lights on at 07.00 and food and water provided ad libitum. All the experiments were conducted according to NIH guidelines and were approved by the Institutional Animal Care & Use Committee (IACUC) of the Louisiana State University School of Veterinary Medicine.

## 2.2. Behavioural test battery

All behavioural experiments were performed by the same investigator during the late morning. One set of experiments was performed per day over a 16-day period. Experiments were performed in the order indicated in table 1.

**Table 1.** Timeline with the details of sequence of behavioural tests done in different strains. Note that the same sequence was followed for all the strains included in this study. With markings depicting test days and empty boxes showing no-test (resting) days.

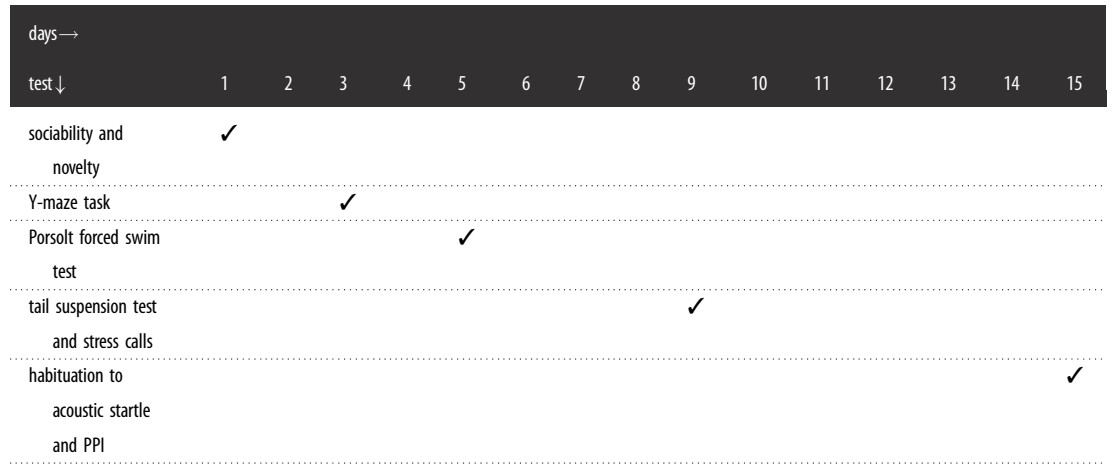

| days→ test↓ | 1 | 2 | 3 | 4 | 5 | 6 | 7 | 8 | 9 | 10 | 11 | 12 | 13 | 14 | 15 |
|---|---|---|---|---|---|---|---|---|---|---|---|---|---|---|---|
| sociability and novelty | ✓ | | | | | | | | | | | | | | |
| Y-maze task | | | ✓ | | | | | | | | | | | | |
| Porsolt forced swim test | | | | | ✓ | | | | | | | | | | |
| tail suspension test and stress calls | | | | | | | | | ✓ | | | | | | |
| habituation to acoustic startle and PPI | | | | | | | | | | | | | | | ✓ |

## 2.3. Social interaction test

Social interaction and social novelty were tested as previously described [39]. Briefly, one day (24 h) prior to the commencement of testing, mice were habituated in the testing room for 30 min to 1 h. The animals were allowed to spend 5 min in the test compartment, 55 × 45 cm (shown in figure 1a), during the habituation phase. Two smaller holding enclosures in the test compartment were designated as 'E (Empty)' in the subsequent habituation phase (5 min). Prior to each phase of the test, the compartment was wiped with 70% isopropyl alcohol to prevent odour specific cues and bias in the subsequent steps. For the sociability test, a strange mouse (S1-stranger 1) was introduced into one of the compartments, following which the test animal was re-introduced into the chamber (5 min). After an inter-trial interval of 30 min, the chamber was wiped clean with isopropyl alcohol, following which a second stranger mouse (S2) was introduced, along with the first stranger (S1). The time in contact with S1 and S2 was measured to determine social novelty. The time spent in contact with E, S1 or S2 in sociability and social novelty tests was determined blindly using ANY-maze software (ANY-maze, Wood Dale, IL). The results were interpreted in terms of per cent of total time spent with S1 i.e. (S1/S1 + E)*100 and S2 i.e. (S2/S1 + S2)*100 for sociability and social novelty respectively.

In addition, this test was used as a measure of anxiety in terms of thigmotaxis, i.e. time spent near the periphery of the chamber [40–42]. During the habituation phase, exploratory behaviour was measured in terms of number of entries into different regions in the chamber (figure 1d). Total distance travelled (figure 1e) in the apparatus was used as a measure of overall activity [43].

## 2.4. Faecal boli count during interaction test

To assess anxiety in terms of the animal's physiological response, a count of total number of faecal boli produced while in the social-interaction chamber was recorded.

## 2.5. Modified Porsolt forced swimming test [44]

As a metric of despair, we used the modified Porsolt forced swimming test (FST) [45]. Animals were placed in a 3 l beaker containing 2.5 l water for a period of 6 min. The camera (1080 HD, Logitech, Newark, CA) was positioned with a side view of the beaker to determine leg movements of the animal. Two animals obscured from each other's view were recorded simultaneously. Scoring of the movements were done according to the method of Can et al. [44]. Per cent mobility time was calculated from a total 4 min testing period, following an initial 2 min acclimation period, which was excluded from this calculation.

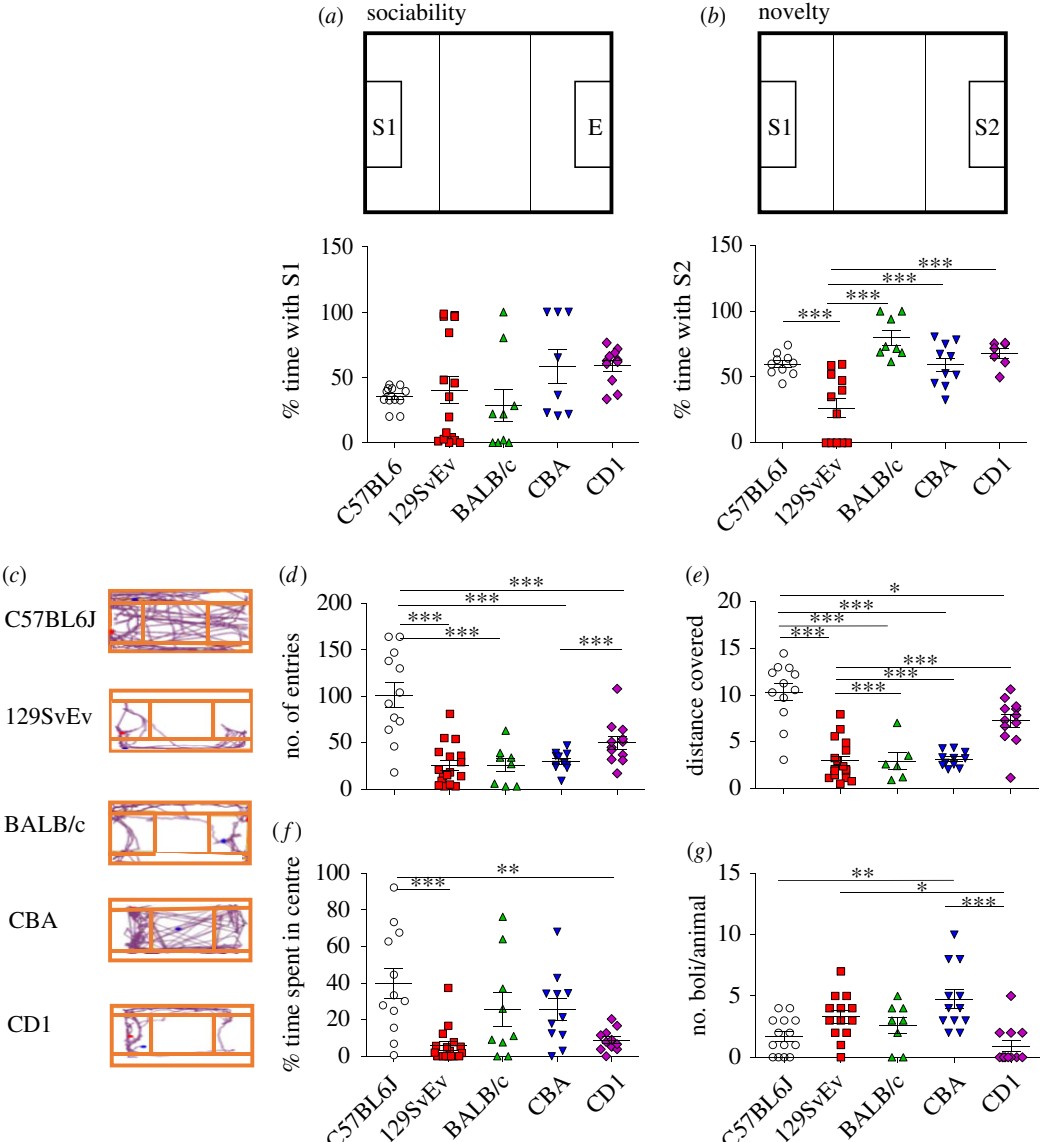

**Figure 1.** Open field test. (*a*) Sociability: fraction of time spent with stranger 1 (S1). (*b*) Novelty: fraction of time spent with stranger 2 (S2). (*c*) Representative track plots in the testing chamber for each strain. (*d*) Exploratory behaviour: total number of entries into different partitions of the apparatus. (*e*) Overall activity: total distance travelled throughout the apparatus. (*f,g*) Measures of anxiety: total time spent in the central chamber and number of faecal boli counted throughout the testing period, respectively. Data shown are calculated at level of significance with $*p \leq 0.05$, $**p \leq 0.01$ and $***p \leq 0.001$.

## 2.6. Tail suspension test

As another metric for despair, animals were suspended by the tail from a custom holder and their mobility during suspension was assessed [27]. The total test duration was 6 min. To remove bias from the initial acclimation period, the first 2 min were excluded from analysis. Per cent mobility time during the latter 4 min test period was calculated. The results were later compared with the per cent mobility in the forced swimming test.

## 2.7. Stress calls

When stressed, animals elicit a unique pattern of ultrasonic vocalizations, which provides an index of the affective state of the animal [46–48]. During the tail-suspension test, stress calls were recorded with an AT125 bat call recorder (Binary Acoustics, Carlisle, PA) and digitally recorded using SPECT'R software (Binary Acoustics). Calls were analysed off-line using SCAN'R software (Binary Acoustics). Calls above

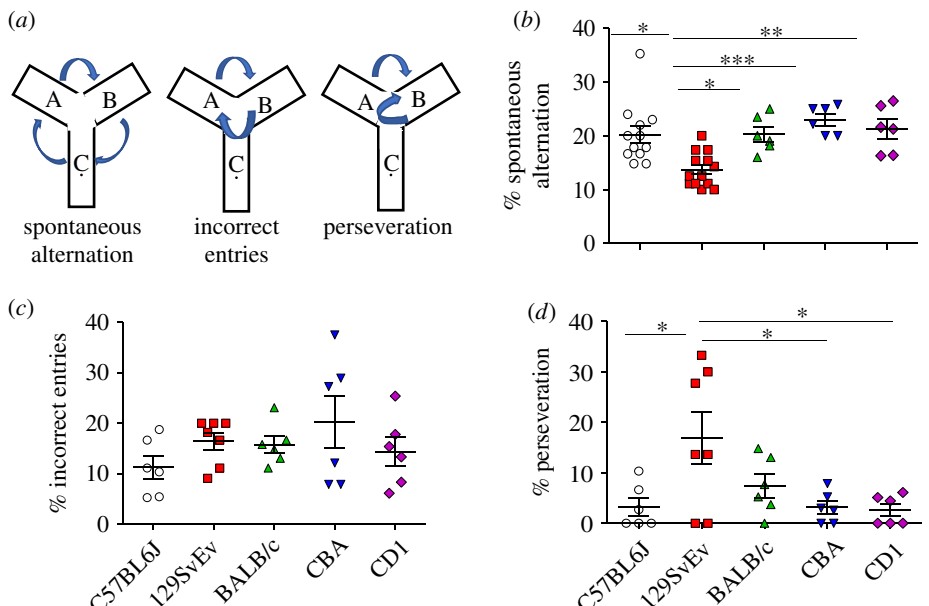

**Figure 2.** Y-maze task. (*a*) Representation of tasks for fig. *b–d*. (*b*) Spontaneous alternation: per cent spontaneous alternation in all the strains. (*c*) Incorrect entries: per cent incorrect entries showing a repetitive behaviour, (*d*) Perseveration: per cent perseveration depicting the failure to shift decision making. Data shown are calculated at level of significance with $*p \leq 0.05$, $**p \leq 0.01$ and $***p \leq 0.001$.

30 kHz (typical of adult mice) with a minimum duration of 5 ms were considered. Number of calls per 6 min were calculated along with the mean and maximum frequency and the duration of each call. We also determined the number of animals producing stress calls during the test. The data were expressed as mean $\pm$ s.e.

## 2.8. Y-maze test

Based on the natural exploratory instinct of rodents [49], the Y-maze task was used to assess spatial short-term memory, as described by Garcia & Esquivel [50]. The Y-maze was custom made from a wooden platform, which has three arms of equal length (35 cm), width (5 cm) and height (10 cm), connected at the centre at an angle of 120° (figure 2*a*). As above, 12 animals from each strain were divided into two groups to assess spontaneous alternations and working memory respectively. The videos were recorded using a Logitech HD 1180 camera and later analysed with ANY-maze or manually.

For assessment of spontaneous alternations, the animal was placed in the centre of the Y-maze; all the arms were opened, and animals were allowed to roam for 7 min to test their exploratory and alternation behaviour. The alternations were scored manually, where a sequence of arm visits (e.g. A to B to C) without repetition was counted as one complete alternation. Incorrect entries occurred when the animal repeated a visit to the first arm after a second arm visit (e.g. A to B to A) and perseveration was scored when the animal visited the same arm again without alternating to the next one (e.g. A to A). All the calculations were normalized as per cent of the total number of entries each animal made to all the arms, as follows [50]:

$$\left( \frac{\text{Alternations}}{\text{total number of possible alternations}} \right) \times 100.$$

Total number of possible alternations = Total of entries − 2.

## 2.9. Y-maze task (as spatial memory task)

As a test of reference and working memory [51,52], the animal was allowed to first explore the Y-maze with arm C closed, while arms A and B were open for exploration. For the second round, with an inter-trial interval of 30 min, animals were placed in the Y-maze for 7 min with all three arms open for exploration. Number of entries, time spent in the previously unexplored arm, as well as those

previously explored, were determined using ANY-maze. Per cent time spent in the correct (i.e. previously closed) arm was calculated as follows:

$$\text{Percentage of time spent on correct arm (\%)} = \left( \frac{\text{Time spent visiting correct arm}}{\text{Total time to visit the three arms}} \right) \times 100.$$

## 2.10. Habituation to acoustic startle and pre-pulse inhibition

Responses to acoustic startle stimuli were used to measure habituation and pre-pulse inhibition (PPI), following the protocol described by Valsamis & Schmid [53]. A custom plexiglass skeleton with adjustable metal rods was used to constrain the animal comfortably, yet snugly, without impeding sound delivery. The chamber was fitted with a piezoelectric pad (Karlsson Robotics, Tequesta, FL) on which the animal stood to measure startle movement. Sound stimuli were delivered through free-field speakers. The total test had two blocks running sequentially. The first block consisted of a series of 15 startle stimuli (white noise: 105 dB; 20 ms), followed by a second block, which contained pre-pulse (75 dB and 85 dB; 4 ms) and pulse (105 dB; 20 ms) stimuli separated by periods of 30 and 100 ms arranged randomly [53]. Each animal was given enough time in the chamber to acclimate (marked by negligible movement and cessation of defecation or urination) with background 70 dB white noise. For stimulus delivery and recording of the startle signal, Audacity software 2.2.2 was used. The startle data were exported into Excel (Microsoft, Redmond, WA) using Python. Further analysis was done using Excel followed by statistical analysis with GraphPad Prism 5 (La Jolla, CA). The data were expressed as mean $\pm$ s.e.

## 2.11. Statistical analysis

ANOVA followed by Tukey *post hoc* test for multiple comparisons was used to determine significant differences among groups. Animals of each gender were analysed separately, but no difference was observed on these tests. As such results are presented with genders combined. For sociability testing, non-parametric test (Kruskal–Wallis test) was used due to non-Gaussian distribution. All the data were expressed as mean $\pm$ s.e. A $p$-value $< 0.05$ was considered statistically significant. In addition, a correlation analysis between related test parameters (e.g. tail suspension test versus forced swim test and anxiety-related behaviours) was done and is discussed in the results section. Statistical analysis was performed using GraphPad Prism 5 (GraphPad Software, La Jolla, CA). The respective number of animals are described in the individual figure legends and Material and methods section of this paper.

## 2.12. Principal component analysis

PCA was performed on data recorded for different tests on test animals using Scikit-learn [54]. PCA was performed with $n = 6$ for all the strains and tests. For all test results to have equal weight, we first standardized the recorded values for each test results to have zero mean and unit variance. On the standardized data, PCA was performed to calculate the values for the first three principal components. The first two/three principal components were plotted to observe the clustering of different test groups.

# 3. Results

## 3.1. Social interaction task

### 3.1.1. Sociability and novelty

As a measure of sociability, we measured the per cent time spent with a stranger mouse (S1), as a function of the total time spent between S1 and the empty chamber (E). The per cent time spent with S1 did not differ significantly across all strains considered in this study. As illustrated in figure 1a, the 129SvEv and CBA/J strains were distributed bimodally. For the 129SvEv strain, 30% of mice exhibited greater sociability, i.e. time with S1 > E. Conversely, 50% demonstrated low sociability, i.e. time spent with S1 < E. Interestingly, C57BL/6J, BALB/c and CD1 mice exhibited less variation in behaviour for mice within the same group.

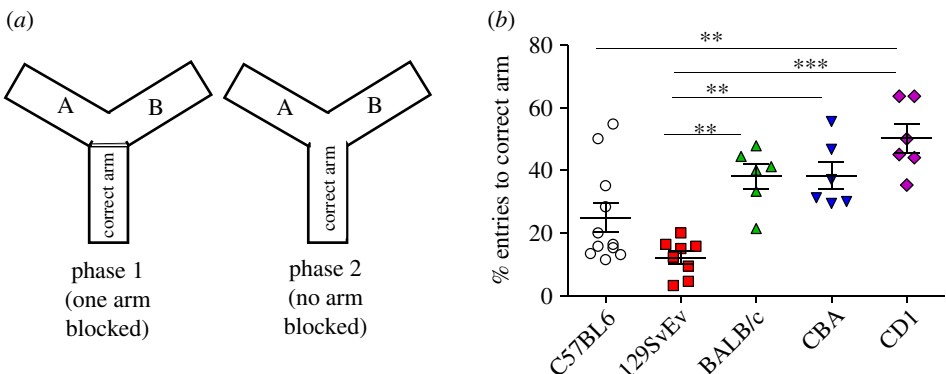

**Figure 3.** Y-maze task with one arm blocked. (*a*) Representation phases of task. (*b*) Per cent entries into the correct arm as a measure of spatial learning and memory. Data shown are calculated at level of significance with $*p \leq 0.05$, $**p \leq 0.01$ and $***p \leq 0.001$.

In the social novelty task, the test mouse was exposed to a second stranger mouse (S2), along with the now familiar conspecific S1 (figure 1*b*). All the strains showed a greater affinity towards the novel stranger mouse (S2), but the 129SvEv spent significantly less time with S2, when compared with C57BL/6J, BALB/c, CBA and CD1 ($p < 0.001$).

## 3.2. Exploratory behaviour and activity

C57BL/6J mice exhibited significant exploratory behaviour when assessed in an open field. Representative track plots in figure 1*c* demonstrate the area and distance covered by mice in the open field arena. Further analysis showed that C57BL/6J mice exhibited a significantly greater number of entries into the three chambers of a sociability box when compared with BALB/c, CBA/J, 129SvEv and CD1 (figure 1*d*; $p < 0.001$). Furthermore, the C57BL/6J exhibited a significantly higher total distance travelled across the three chambers (figure 1*e*); BALB/c, CBA/J, 129SvEv ($p < 0.001$), and CD1 ($p < 0.05$) as calculated using one-way ANOVA followed by Tukey's *post hoc* test.

## 3.3. Thigmotaxis and faecal boli production

Thigmotaxis was determined by the per cent time spent in the periphery of the open field arena. As shown in figure 1*f*, C57BL/6J mice spent significantly greater time in the centre of the apparatus than 129SvEv and CD1 mice ($p \leq 0.001$ and $p \leq 0.01$ respectively). All other strains spent similar amounts of time in the periphery as C57BL/6J, but were not significantly different from 129SvEv and CD1.

Faecal boli production was also analysed (figure 1*g*) and was found to vary among strains, with the greatest number produced by CBA, suggesting high anxiety. The number of faecal boli was greater in CBA than C57BL/6J and CD1 ($p \leq 0.01$ and $p \leq 0.001$ respectively) and least by C57BL/6J.

## 3.4. Spontaneous alternation in Y-maze

Animals were selected for analysis based on a cut-off of 10 entries to the arms (figure 2*a*), two animals from 129SvEv, one BALB/c, one CBA and one CD1 mice were not included in the analysis because of fewer entries than the required threshold. The number of spontaneous alternations were significantly lower in 129SvEv than C57BL/6J ($p \leq 0.05$), BALB/c ($p \leq 0.05$), CBA ($p \leq 0.001$) and CD1 ($p \leq 0.01$). All other strains were not significantly different from one another. When calculating per cent incorrect entries and perseveration (figure 2*c,d*), we found 129SvEv mice also showed greater perseveration than C57BL/6J, CBA and CD1 ($p \leq 0.05$). Interestingly, the per cent of incorrect entries was not significantly different in any of these groups as determined by one-way ANOVA followed by Tukey *post hoc* test.

## 3.5. Working memory task in Y-maze (one arm blocked)

In the Y-maze task for spatial memory (figure 3), BALB/c, CBA and CD1 mice spent significantly more time in the correct arm ($p \leq 0.001$) than 129SvEv. However, no significant difference was observed

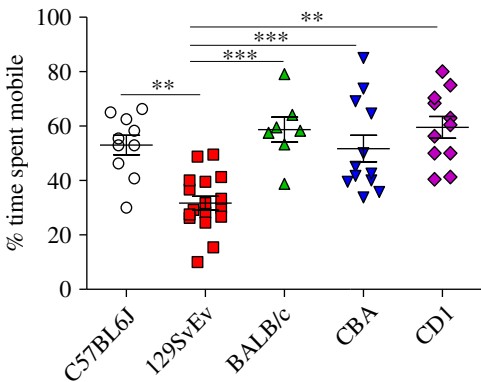

**Figure 4.** Porsolt forced swim test: per cent time spent mobile during forced swim test as a measure of despair in all the mentioned strains. Data shown are calculated at level of significance with *$p \leq 0.05$, **$p \leq 0.01$ and ***$p \leq 0.001$.

between C57BL/6J and 129SvEv. We also found no difference among BALB/c, CBA and CD1 mice. The per cent number of entries to the correct arm by all the strains considered did not show any significant difference (data not shown). This test was conducted with a separate group of animals from the spontaneous alternation group.

## 3.6. Porsolt forced swim test

This test measured coping behaviour (or despair) during stressful conditions (figure 4). Based on a measure of per cent mobility over a period of time, the 129SvEv strain exhibited lower mobility time; versus other strains: C57BL/6J ($p \leq 0.01$), CD1 ($p \leq 0.01$), CBA ($p \leq 0.001$) and BALB/c ($p \leq 0.001$). Although indicative of despair-like behaviour, 129SvEv strains may be generally inactive, as noted above, which could potentially confound this as a direct despair measure for this strain. When we compared C57BL/6J, CD1, CBA and BALB/c, there was no observable difference in the per cent mobility time.

## 3.7. Tail suspension test

The tail suspension test (TST) was also used to assess despair-related behaviour in animals under challenging situations. Similar to the forced swim test, the 129SvEv mice exhibited a significantly reduced mobility time when compared with all other strains. The per cent mobility time for 129SvEv mice was reduced significantly when compared with CD1 ($p \leq 0.001$), C57BL/6J ($p \leq 0.05$) and BALB/c ($p \leq 0.01$) whereas no significance difference was found versus CBA (figure 5a).

## 3.8. Stress calls

Ultrasonic vocalizations are produced by mice while showing aggression, stress or mating behaviour towards the same or opposite gender. Call behaviours are also a common metric for neurodegenerative disorders [55,56]. In our study we found that C57BL/6J produced significantly greater number of calls during the tail suspension test than 129SvEv ($p \leq 0.001$), CBA and CD1 ($p \leq 0.05$) as shown in figure 5b. The duration of calls also differed, where C57BL/6J had significantly longer duration calls than 129SvEv ($p \leq 0.001$), CBA ($p \leq 0.01$) and CD1 ($p \leq 0.05$) (figure 5c,d). BALB/c differed from 129SvEv ($p \leq 0.01$) and CD1 ($p \leq 0.05$). We found no difference in the call duration between C57BL/6J, BALB/c and CBA. The number of animals that vocalized varied in each group, with BALB/c having the lowest number of animals producing calls (8%), followed by 129SvEv (16.7%), CBA, C57BL/6J (58.8% each), and CD1 ranking the highest (66.7%). Maximum and mean frequency of calls did not differ significantly. Stress calls did not correlate with anxiety measures (highest in 129SvEv versus all other strains), probably due to the threshold criteria selected for counting the calls (intensity >30 kHz and minimum duration >5 ms).

## 3.9. Habituation to acoustic startle (AS) and pre-pulse inhibition

Animals were tested to habituation of the acoustic startle response (ASR) to 20 ms pulses of 105 dB white noise [53]. All the strains eventually habituated to the acoustic stimulus (figure 6a). The strains varied, with

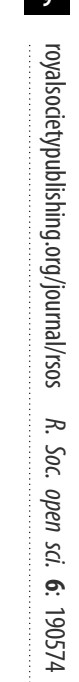

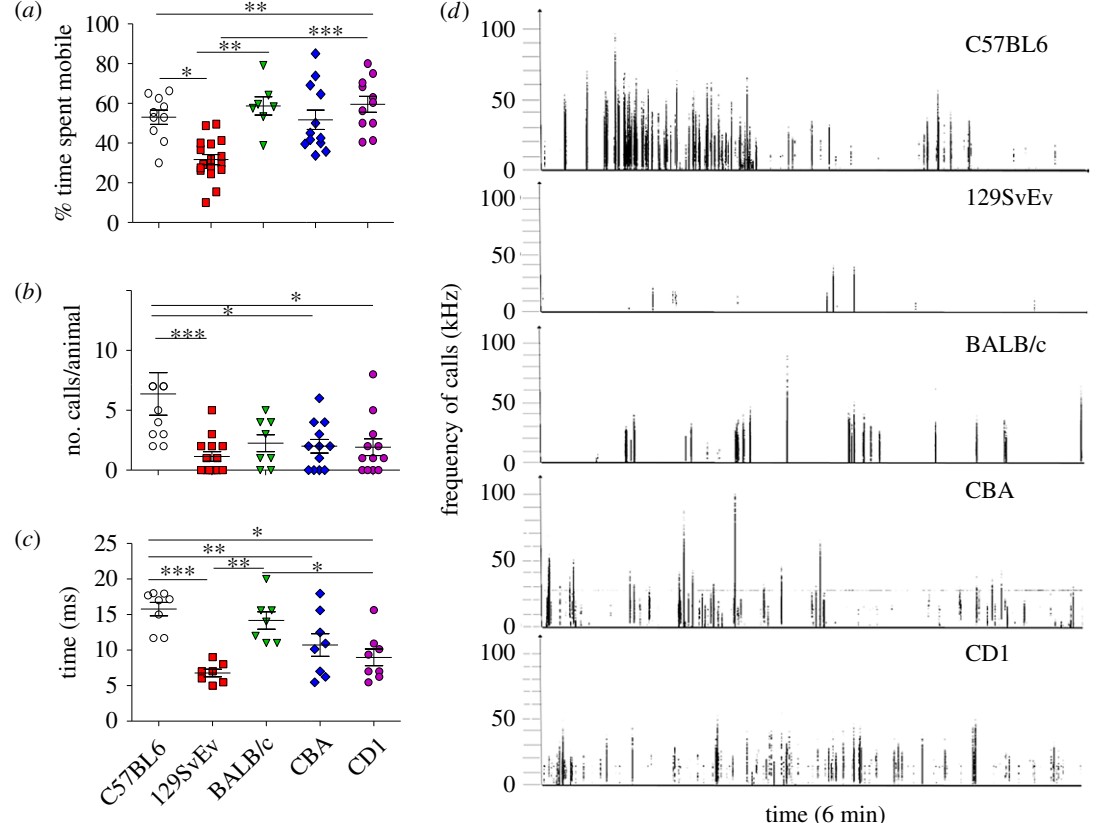

**Figure 5.** Tail suspension test and stress calls. (*a*) Mobility time: per cent time spent mobile during tail suspension test. (*b*) Number of stress calls: number of stress calls in the mentioned strains as recorded by ultrasound recorder. (*c*) Duration of stress calls: duration of calls in different strains in ms. (*d*) Representative spectrograms of calls produced by each strain in a 6 min period. Data shown are calculated at level of significance with *$p \leq 0.05$, **$p \leq 0.01$ and ***$p \leq 0.001$.

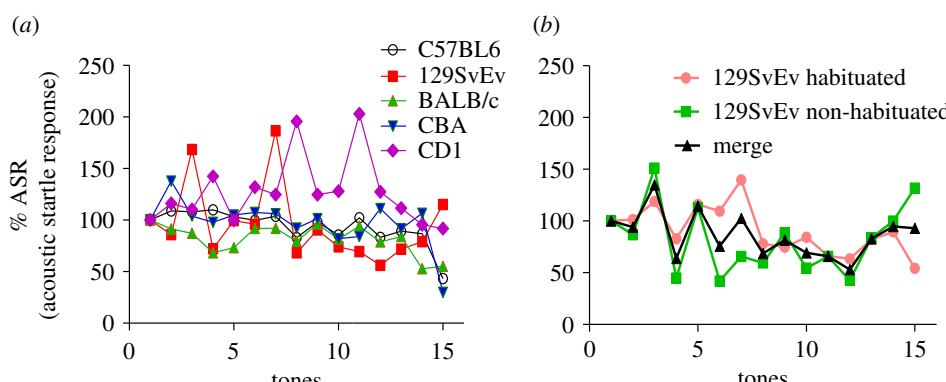

**Figure 6.** Acoustic startle response. (*a*) Habituation: per cent habituation during 105 dB white noise delivery. (*b*) Differential habituation in 129SvEv: two categories of 129S, habituating (pink), non-habituating (green) and merge (black) of all the animals.

70% habituation in C57BL/6J, 66% in CBA and 50% in BALB/c followed by 7 and 11% in 129SvEv and CD1, respectively. In addition, interestingly 129SvEv animals showed a bimodal behaviour in terms of habituation, some animals habituated well whereas others did not, as shown in figure 6*b*.

The second block of stimuli were used to assess pre-pulse inhibition (figure 7). Stimuli consisted of combinations of variable tone intensities (75 or 85 dB) separated from the startle stimulus by inter-stimulus intervals (ISI) of 30 or 100 ms at random [53]. At 30 ms and 75 dB pairing, there was a significant difference between CD1 versus CBA ($p \leq 0.001$), C57BL/6J and CBA ($p \leq 0.05$), CD1 versus BALB/c ($p \leq 0.05$). Only CBA and CD1 differed significantly ($p \leq 0.05$) at 100 ms and 75 dB.

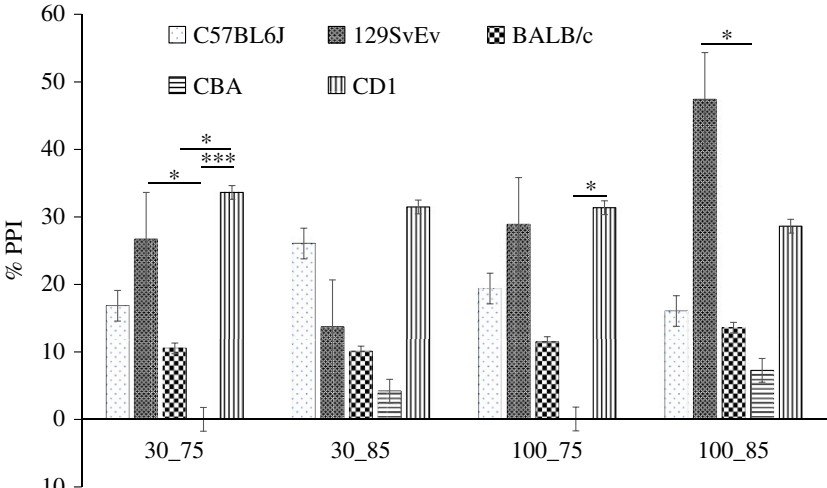

**Figure 7.** Pre-pulse inhibition: pre-pulse inhibition in different strains at different tones and inter-stimulus intervals, i.e. at 30_75, 30_85, 100_75 and 100_85 respectively where x-axis represents duration (30 versus 100 ms) and intensity (75 versus 85 dB) and y-axis depicts per cent habituation. Data shown are calculated at level of significance with *$p \leq 0.05$, **$p \leq 0.01$ and ***$p \leq 0.001$.

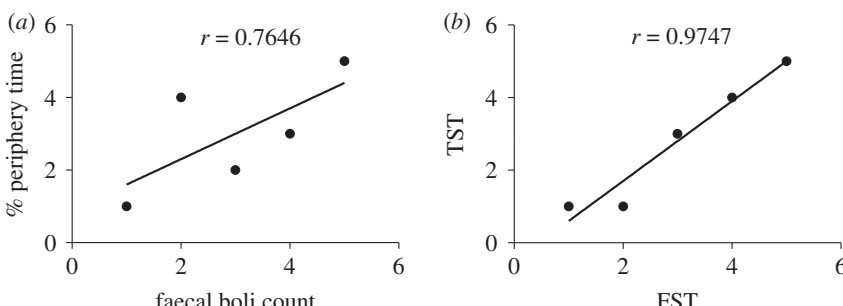

**Figure 8.** Correlation coefficient: (a) correlation coefficient between time spent at periphery and faecal boli count, (b) correlation coefficient between tail suspension test and forced swimming test.

At 30 ms and 85 dB pairing there was no significant difference between the strains, whereas the CBA strain showed significantly lower PPI than 129SvEv ($p \leq 0.01$) at 100 ms and 85 dB.

## 3.10. Correlation of anxiety- and despair-related behaviours

Anxiety-related measures (faecal boli production and time spent in the open field) were positively correlated with one another; Spearman's rank order correlation exhibited $r = 0.7646$, indicating a moderate positive but non-significant ($p > 0.05$) correlation (figure 8a). When individual groups were analysed with each animal as a single data point, we did not see a significant correlation between the two variables in any of the strains, except for BALB/c (Pearson's $r = 0.8471$ and $r^2 = 0.7175$, $p \leq 0.05$). In addition, we calculated the correlation between despair-related behaviours (the immobility time during forced swim test and tail suspension test) and found a positive correlation between the two variables (Spearman $r = 0.9747$, $p \leq 0.05$) (figure 8b).

## 3.11. Principal component analysis

Figure 9 shows PCA for behavioural assays, which included all the tests performed in the present study. Remarkably, $n = 6$ mice from each line completed all assays, with 129SvEv and C57BL/6J strains forming clearly differentiated clusters from each other as well as the other strains (BALB/c, CBA, CD1) tested.

# 4. Discussion

This study compared performance on common behavioural tests among the C57BL/6J, CBA, BALB/c, 129SvEv (inbred lines) and the CD1 (outbred line) mouse strains. Our findings demonstrate

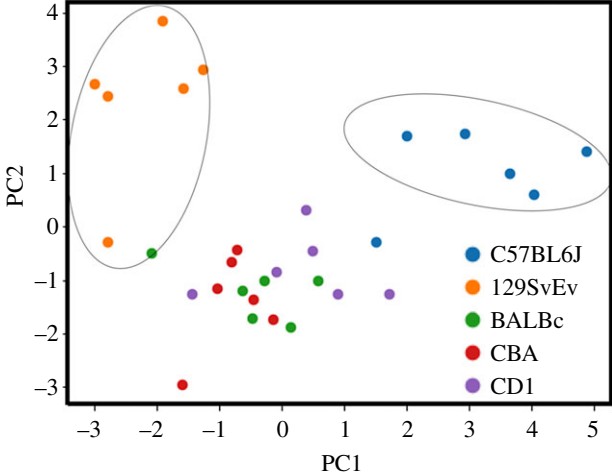

**Figure 9.** Principal component analysis (PCA) of all the tests. The graph depicts PCA for $n = 6$, with all tests included.

**Table 2.** Summary table depicting overall results for all the tests in different strains. Each row/column intersection denotes the behaviours that were statistically different ($p \leq 0.05$) between strains.

| strains | C57BL6 | 129SvEv | BALB/c | CBA | CD1 | |
|---|---|---|---|---|---|---|
| | | | | | | 1. sociability |
| | | | | | | 2. novelty |
| C57BL6 | | | | | | 3. exploratory behaviour |
| | | | | | | 4. overall activity |
| 129SvEv | 2,3,4,5,7,9,11,12,13 | | | | | 5. thigmotaxis |
| | | | | | | 6. defecation |
| | | | | | | 7. spontaneous alternation |
| BALB/c | 3,4 | 2,10,11,12,15(100_85) | | | | 8. incorrect entries |
| | | | | | | 9. perseveration |
| | | | | | | 10. Y maze |
| CBA | 3,4,6,13,15(30_75) | 2,7,9,10,11,12 | none | | | 11. FST |
| | | | | | | 12. TST |
| CD1 | 3,4,5,10,13 | 2,4,6,9,10,11,12 | 2,15(30_75) | 4,6,15(30_75,10_75) | | 13. stress calls |
| | | | | | | 14. habituation |
| | | | | | | 15. PPI |

differences in behaviour among these strains, which support the notion that background genotype is an important factor in evaluating mouse models of neurological disorders in biomedical research [27,28,56,57]. This is reported even for observations in their home cages, since strains with different genetic backgrounds exhibit a wide range of variation in their sheltering behaviour [58–60]. The present study was divided into various time points to carry out each of the listed tasks in the same set of animals (table 1).

These mice strains showed both similarities (social interaction, novel recognition, FST, TST, spatial memory, Y-maze tasks, etc.) and differences (thigmotaxis, stress calls, habituation to acoustic startle, etc.) in behavioural test performance, more clearly specified in the summary table (table 2) and PCA graph (figure 9). Our data support prior studies comparing phenotypic characteristics of BALB/c with C57BL/6J and 129SvEv, as possible models to study mental health disorders [13,34,35,61]. Some discrepancies with prior studies, such as more time spent by C57BL/6J with S1 [31] and BALB/c having the least locomotor activity when compared with other strains [62], could be explained by variations in the age, gender of animals, parameters of interaction and disparity in the apparatuses used [63–65]. For instance, C57BL/6J exhibited the most exploratory behaviours of all the strains

tested here, similar to that observed by Dellu *et al*. [66] for distance covered inside the apparatus. However, the 129SvEv strain exhibited the most behavioural differences from other strains (table 2, figure 9). Furthermore, habituation to acoustic startle and PPI showed variation among the inbred and outbred strains. These data corroborate many studies where inbred strains were compared for characteristic startle [12,67,68] and electrophysiological responses [69], with observed differences attributed to genetic background.

Naturally, the genetic underpinnings of these observed behavioural differences are complex, intertwined, and being actively investigated. Certain candidate genes may play a particularly important role in some behaviours. For instance, genetic and/or regional neuroanatomical expression differences in the number of nicotinic receptor binding sites [70], which varies among strains [71], are probably involved in PPI of the acoustic startle response (ASR) [72,73]. Similarly, difference in habituation responses could be explained by variable thresholds for acoustic startle among strains, mediated by these receptors [74]. Interestingly, Hazlett *et al*. [75,76] noted that the PPI differences are an important indicator of schizophrenia and schizotypic disorders in humans and animal models of the disease, including rodents [77]. Relative differences among animals in the strength of PPI may also be due, in part, to changes in auditory sensitivity [78,79], which is known to progress at different rates in various strains, e.g. C57BL/6J and CBA/J exhibit early sensorineural hearing loss [80], while other strains hear normally when young, so PPI and ASR differences may be due to central auditory processes. The differences in the levels of habituation to ASR and PPI in these inbred strains can thus facilitate studies of the genetic basis of sensorimotor gating.

Interestingly, the 129SvEv strain exhibited the most clearly differentiated results in all the tests compared with the other strains (table 2 and figure 9). Although this strain differs from the other strains at many genetic loci, as mentioned before, it is interesting to note that it, along with all other 129s strains, have a spontaneous mutation at exon 6 of the *Disc1* (Disrupted in Schizophrenia) gene (coding a scaffolding protein called DISC1) leading to a C-terminal truncation, affecting its functionality [34,81]. To the extent that the 129SvEv strain harbours such a schizophrenia-related gene mutation, it is noteworthy that its behaviours were consistent with those found in the manifestation of the disease in humans [33]. For instance, our study shows reduced performance of tasks associated with the prefrontal cortex and the hippocampus (sociability, novelty and Y-maze for spatial memory, respectively), similar to that observed by others [21,22,34,82]. In addition, this strain exhibited a bimodal distribution in many behaviours, e.g. sociability, perseveration, number of calls and habituation to acoustic startle, etc., which we speculate mimics similar phenotypic variations in humans with the genetic mutation [83–87]. Interestingly, the 129SvEv strain exhibits neuroanatomical variation of the corpus callosum, lateral ventricles and cortical structure [70,88,89], which may influence its behavioural phenotype, although it remains to be determined whether these relate to the tested behaviours specifically. These behavioural findings also support the use of the 129SvEv strain as a genetic model of psychotic disorders such as schizophrenia, bipolar disorder, schizophrenia-like disorders, linked to the *Disc1* gene [33,90–92], and could be employed to address the distribution and regulation of the molecular cascades responsible for such a unique phenotype of this strain from those with different genetic backgrounds. These results also suggest that some caution may be warranted in interpreting the behavioural effects of genetic mutations introduced into the 129SvEv strain.

In addition, PCA also demonstrated that the C57BL/6J mice formed a separate cluster of behavioural phenotypes, diverging from all other strains (inbred (BALB/c, CBA) or outbred (CD1) lines) of mice that were included in this study, supporting prior behavioural studies on this strain [17,62,93,94]. Indeed, the C57BL/6J strain, despite its ubiquity in biomedical research, has been noted to have numerous unique affectations, obesity, presbycusis, ill-temperament, etc. [95,96]. Like the 129SvEv strain, some caution may be warranted in considering behavioural assays on this strain, particularly given the ubiquity of this strain as a background for many studies [36,57,97].

In conclusion, the present study identified differences in behavioural phenotypes due to genetic background in several mouse strains. These results provide a potential baseline for assessing the genetic basis of behaviour among these strains. Particularly interesting are the phenotypes for C57BL6J and 129SvEv inbred strains and CD1 outbred strain [88,98], which form unique clusters differing from one another as well as the other groups tested (figure 9). By performing batteries of behavioural tests in the same laboratory set-up, our study has attempted to control for the effects of genetic background on behaviour.

It is important to reiterate, however, that the degree to which genetic background influences introduced mutations remains one that must be addressed empirically. Indeed, mutations introduced onto different background may have different outcomes in terms of absolute values of effects between

strain, while the magnitude of the differential effect size, in terms of standard deviations between mutants and wild-types may in fact be similar for transgenes introduced among strains [11,20]. Therefore, as with most considerations of experimental biological parameters, we suggest that the treatment of genetic background in the assessment of mouse models of neurological disorders be regarded as one of degree, not that of hegemony.

Ethics. All the experiments were conducted according to NIH guidelines and were approved by the Institutional Animal Care & Use Committee (IACUC) of the Louisiana State University School of Veterinary Medicine.

Data accessibility. The datasets supporting this article and data for analyses in Graph pad have been uploaded as part of the electronic supplementary material.

Authors' contributions. R.S. and C.C.L. contributed to the conception and design of the experiments, analysis, interpretation of data and drafted the article or revised it for important intellectual content. O.M.O. contributed to experiments, analysis and interpretation of data. All authors approved the final version of the manuscript.

Competing interests. The author states that the present manuscript presents no conflict of interest.

Funding. This study was supported by NIH grant nos. R03 MH 104852 and NSF IOS 1652432 awarded to C.C.L.

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
