## [Reviewer comments · Royal Society Open Science]

Review History

RSOS-181399.R0 (Original submission)

Review form: Reviewer 1 (Gidon Felsen)

Is the manuscript scientifically sound in its present form?

Yes

Are the interpretations and conclusions justified by the results?

Yes

Is the language acceptable?

No

Is it clear how to access all supporting data?

Yes

Do you have any ethical concerns with this paper?

No

Have you any concerns about statistical analyses in this paper?

No

Recommendation?

Major revision is needed (please make suggestions in comments)

Comments to the Author(s)

Sultana and colleagues compare performance on standard behavioral tasks between several commonly-used mouse strains. They show some significant differences in behavior, demonstrating that caution is warranted when interpreting comparative data collected in multiple strains. The manuscript is fairly straightforward and the quantification of differences in behavior across strains could provide a valuable resource to the community. However, there are several concerns about the manuscript as currently written:

-The authors make it sound as if researchers often attempt compare results across strains. This would clearly be a problem, but how common is it? Does it occur in primary research studies, and/or in reviews attempting to synthesize the literature? Citing some papers in which this sort of problematic comparison is made, and how that has negatively affected the papers' conclusions, would bolster the rationale for the current manuscript.

-Is it possible that behavior was affected by the order in which the tests was performed? Keeping the order consistent across strains is appropriate, but perhaps the results would have been different had the experiments been conducted in a different order. Performing a new set of experiments using a different order could address this possibility; alternatively, it may also be possible to address it in the discussion section.

-Some strains appear to show much more (or less) variability than others on particular behavioral tests. It might be valuable to quantify this variability, as it could indicate how to interpret the results of a given test for a particular strain. For example, it seems notable that in Fig. 1A, C57BL6 mice exhibit much lower variability than, e.g., 129SvEv mice.

-The figure and table legends are missing.

-What is shown on the axes of Fig. 5D?

-It is not clear why the color scheme in Fig. 6A differs from that in the preceding figures.

-Fig. 8 seems out of order and isn't referred to in the Results section. Also, it is not clear whether there is any significance to the markers being colored red in this figure. Does this correspond to the color scheme in other figures?

-There are numerous spelling errors, particularly in the Results section. Just as one example, p. 11 line 3 (significnatly, distnace, accross).

Gidon Felsen

Department of Physiology and Biophysics

University of Colorado School of Medicine

(I am including my name/affiliation because I have opted in to open peer review)

Review form: Reviewer 2

Is the manuscript scientifically sound in its present form?

No

Are the interpretations and conclusions justified by the results?

No

Is the language acceptable?

No

Is it clear how to access all supporting data?

Yes

Do you have any ethical concerns with this paper?

No

Have you any concerns about statistical analyses in this paper?

Yes

Recommendation?

Reject

Comments to the Author(s)

Comments here indexed by [page].[line]

3.23 "...few inter-strain comparisons are available." This is not really true in general. Perhaps you need to be more specific? Unfortunately, failure to acknowledge the vastness of the literature (or postponement to the Discussion as here) immediately raises a question in any reader's mind about the value of your work. I am not saying your work is not valuable, but failure to put it in the context of the large available literature deprives your work of much of its potential value in the mind of the reader. In case you find that your work is largely not original, that is not a real problem, since - and I think this is an important and useful aspect of journals like RSOS - "studies testing the reproducibility of significant work are encouraged". You just need to be open about this and highlight what part of your work is new and what is confirmatory. In any case, the four references provided are only a small subset of the literature available, which addresses the problem of background variation on two levels. First, many papers, stretching back many years, address various aspects of genetic background and its contribution to differences in wildtype behaviour. Second, beyond the issue of wildtype comparisons, there are reports of interactions of induced mutations with genetic background. Many of these are anecdotal (single gene or behaviour), but a couple of recent systematic studies were done, one by Sittig et al (many mutants in multiple backgrounds, *Neuron*, 2016) and one by van de Lagemaat et al (three mutants in C57BL/6J vs 129S5, *Genes Brain and Behaviour*, 2017). There are at least ten other references in the latter paper looking at one or the other aspect of genetic background in mouse behaviour. I also note you have more papers referenced further along in your manuscript.

5.28 boli? Call them fecal boli

5.33 put ref 19 with ref 20.

6.49 is it alterations or alternations? Please be consistent.

7.44 snuggly is not the same as snugly, the first of many typos.

8.23 Kruskal

9.9 in 7.5 lines of writing, 8 typos. Out of respect for your reviewers, proofreading for spelling, punctuation and grammar are essential!

10.33 second half of the sentence is hard to follow and appears to jump to the conclusion that mice produce faecal boli due to anxiety alone. Better to use the phraseology “observation X suggests interpretation Y” rather than “we observed X because Y”.

10.37 What is the p-value of the correlation in Fig 8A? By looking, I doubt it is significant. A quick check in R tells me the p-value for your rank correlation is 0.2333. You need more data (one data point per animal?) to prove a correlation. Furthermore it is pretty clear that this is a strain effect. Does this correlation (if it exists) also exist (i.e. achieve significance) within a strain?

10.52 129 mice exhibit less alternations than other strains. What might explain that? Could this be because 129s are inactive?

11.3 129 also show greater perseveration. Why? Might this be because 129s are just inactive (Fig 1)?

11.6 How did you determine difficulty of 129 mice in changing arms? Or might it be they just didn't care, or were too anxious? How do you differentiate between these possibilities?

11.18 129 mice often exhibit bimodal activity, very active or very inactive, and this may change within a day. C57BL/6J are much more active. When you then parameterise the performance of an individual mouse by percentage of visits to the correct arm of the Y maze, your measures of 129 mice will vary widely, and this will decrease your power to detect a C57BL/6J-129 difference. You need to understand and lead the reader through this.

11.38, 11.49 general tendency to inactivity in 129 mice is well known and predicts that they will stop swimming or struggling sooner without needing to invoke any “depression”. You should mention this confounding effect and address how significant your results are after accounting quantitatively for this prediction.

12.18 Where did DISC1 (presumably k/o) mice come from? What background? What dataset were you using? Or did you assess a cohort of mice? What methods did you use? What were the controls? What was the hypothesis? Why were DISC1 mice not mentioned before? Where are the relevant statistical methods? How does this fit into the context of the paper?

12.32 it seems surprising that 129 mice should have second-fewest stress calls when literature suggests 129 strains generally are more anxious. Why is that? What confounds might account for it?

13.20 How did you choose your animals for PCA? Choosing animals opens the door to gross bias and forbids any interpretation of these results (unless you can provide an algorithm demonstrating that these choices were made blind). Please remove this section from the paper.

13.43 You appear to jump to the conclusion that because wildtype mice differ in performance in these tasks, mouse models of human disorders (which usually involve some induced mutation) will also be affected. While this is certainly true at the level of gross measures (cm travelled, cm/s speed, etc), the work of van de Lagemaat et al (2017) shows that this may not be the case, even for backgrounds as different as C57BL/6J and 129S5, when one considers the Cohen d effect size of

the mutation. In other words, although the backgrounds may exhibit different baselines and scales for behavioural measures, the behavioural change in mutants relative to wildtypes may nevertheless be in the same direction and with the same number of standard deviations in both strains.

15.10 here you finally mention Disc1. If it is important, move it up to the introduction.

15.26 on what basis do you argue that bimodal behaviour suggests variable penetrance? Is the behaviour mode an immutable characteristic of an individual animal? How did you test this?

15.35 from the previous comment, you go on to say “other factors” than genetic background play a crucial role in developing an animal’s behavioural phenotype. If lower mobility (for example in 129) were actually an immutable characteristic of an individual mouse, you might make this claim. However, you have not tested this.

16.3 Again, because you have chosen a subset of mice for your PCA, the reader should ask if the observed clusters are just an artefact of which animals were chosen. Probably you should remove the PCA from the paper.

16.20 Are the results in this paper a useful baseline for further studies in neurological disorders? People have been saying this for years, but when really carefully tested in light of mutation effect sizes (number of standard deviations between mutant and wildtype) and random sampling, the effect of background strain seems less important. I would not say UN-important, but less so. You should address this in your manuscript too.

Overall conclusions:

Overall, there may be some good data here. It suffers from several major problems and needs a lot of work before it is ready for publication:

- 1) In its initial form, the fact that in the introduction the manuscript largely ignores a huge literature by very big-name individuals over many years is annoying and biases the informed reader against the manuscript in its current form.
- 2) The experienced reviewer is further annoyed by the poor quality of the spelling and grammar (although this is worse in some places than others). Figure 8 is out of order too. These show general sloppiness. To address this, use a good grammar and spell checker and proofread your manuscript (and have others proofread it) before sending it out. Your reputation as a scientist depends in part on this.
- 3) There is a lack of sophistication in your work in that it lacks proper controls/baseline measures (e.g. home cage behaviour).
- 4) There is a general lack of sophistication in your interpretation of the data, leading you to jump to conclusions along the lines of commonly held views. This is also done without citing the relevant papers. To address this, you should, first, read lots of papers in this field and seek to internalise the careful language and arguments of others. It will help you make your own arguments.
- 5) You nowhere mention the sex of the animals. Even if it has no impact on these measures (see your Moy et al reference), it is generally a big issue in mouse models, so you should mention it and mention how you tested/analysed it. If you didn’t test the effect of sex or use only one sex, it greatly undermines your whole experiment.
- 6) Your study performs several assays on wildtype mice from five strains. Unfortunately, your methods leave many questions unanswered. For example, in the sociability test is similar to one done with five strains (a different set of strains) by others with much more finesse (i.e. including olfactory tests and comparing to home cage behaviour) 14 years ago. That study was cited by you (Moy et al, 2004). With time, we expect methods to improve, but this is not the case here. I

think you have made a good start, but you need a lot more controls and explanation before this paper is ready for review or publication.

Review form: Reviewer 3

Is the manuscript scientifically sound in its present form?

Yes

Are the interpretations and conclusions justified by the results?

Yes

Is the language acceptable?

Yes

Is it clear how to access all supporting data?

Yes

Do you have any ethical concerns with this paper?

No

Have you any concerns about statistical analyses in this paper?

No

Recommendation?

Accept with minor revision (please list in comments)

Comments to the Author(s)

In the manuscript the authors aim to assess and draw attention to phenotypic differences in inbred mouse strains. After a series of behavioral assessments conducted over 16 days the authors conclude that there is a large variability in the response of different strains to various behavior tasks and recommending that future experiments including these strains should be aware of these differences prior to and during the interpretation of any results acquired through the use of these mice as disease models.

The results of these studies draw attention to the importance of considering mouse strain when designing your experiments. This concept is touched upon in other publications, but is not always considered by experimenters. These studies are a good reminder of the importance of this and provides some direct comparisons of common mouse strains that can aid experimenters in their selection.

The behavior tasks utilized in this study seem to be carried out well and assessed well.

Experiments also appear to be spread out over a 16 day period and ordered in a manner that places the most stressful experiments towards the end, controlling for the impact of stress external to the desired behavioral experiment impacting the performance of the mice

Minor Criticisms:

-Author should emphasize a little more that any number of differences could be responsible for their behavioral performance, besides just nicotinic binding sites and DISC1

-Authors should bring to attention other published differences found in 129SvEv line, as this is a line they have chosen to emphasize, and how it compares to their results, for example: previous studies regarding fear and extinction

- the use of CD1 as an outbred strain is a good one, but no discussion is made comparing the CD1 to the inbred strains and what that means.

-In the results section the results given by one strain is referred to as “normal” while the behavior may be the anticipated result, the authors should refrain from the use of the word “normal” as behavioral is considered “normal” only based on previous studies that have utilized specific inbred mouse lines themselves (usually B6)

-there are many spelling errors throughout the manuscript that needs to be addressed, particularly with the misspelling of the word “significant” and “significantly” which occurs at several points throughout the manuscript.

Decision letter (RSOS-181399.R0)

19-Oct-2018

Dear Dr Sultana:

Manuscript ID RSOS-181399 entitled "Contrasting characteristic behaviors among common laboratory mouse strains" which you submitted to Royal Society Open Science, has been reviewed. The comments from reviewers are included at the bottom of this letter.

In view of the criticisms of the reviewers, the manuscript has been rejected in its current form. However, a new manuscript may be submitted which takes into consideration these comments.

Please note that resubmitting your manuscript does not guarantee eventual acceptance, and that your resubmission will be subject to peer review before a decision is made.

Your resubmitted manuscript should be submitted by 18-Apr-2019. If you are unable to submit by this date please contact the Editorial Office.

Please note that Royal Society Open Science will introduce article processing charges for all new submissions received from 1 January 2018. Charges will also apply to papers transferred to Royal Society Open Science from other Royal Society Publishing journals, as well as papers submitted as part of our collaboration with the Royal Society of Chemistry (<http://rsos.royalsocietypublishing.org/chemistry>). If your manuscript is submitted and accepted for publication after 1 Jan 2018, you will be asked to pay the article processing charge, unless you request a waiver and this is approved by Royal Society Publishing. You can find out more about the charges at <http://rsos.royalsocietypublishing.org/page/charges>. Should you have any queries, please contact openscience@royalsociety.org.

Kind regards,
Royal Society Open Science Editorial Office
Royal Society Open Science

on behalf of Dr Ryan Y Wong (Associate Editor) and Prof. Kevin Padian (Subject Editor)
openscience@royalsociety.org

Editor Comments:

Please pay careful attention to the comments of reviewers if you choose to revise and resubmit. In particular please make sure that spelling and grammatical errors are corrected in the next draft, or we will be unable to consider the manuscript further. Thanks for your efforts.

Associate Editor Comments to Author (Dr Ryan Y Wong):

Dear Dr. Sultana,

Your manuscript was reviewed by 3 reviewers. Overall the reviewers agree that the data presented in the manuscript could be of value to the field. However the reviewers have identified a number of concerns, which are included. I agree with the concerns that the current manuscript is not sufficiently placed within the broader context of the field or acknowledges the many studies examining similar questions. I also share the view that the methodology and justification for use of particular behavioral endpoints or data analyses are not clearly described, which makes the data interpretation more difficult.

Reviewers' Comments to Author:

Reviewer: 1

Comments to the Author(s)

Sultana and colleagues compare performance on standard behavioral tasks between several commonly-used mouse strains. They show some significant differences in behavior, demonstrating that caution is warranted when interpreting comparative data collected in multiple strains. The manuscript is fairly straightforward and the quantification of differences in behavior across strains could provide a valuable resource to the community. However, there are several concerns about the manuscript as currently written:

-The authors make it sound as if researchers often attempt compare results across strains. This would clearly be a problem, but how common is it? Does it occur in primary research studies, and/or in reviews attempting to synthesize the literature? Citing some papers in which this sort of problematic comparison is made, and how that has negatively affected the papers' conclusions, would bolster the rationale for the current manuscript.

-Is it possible that behavior was affected by the order in which the tests was performed? Keeping the order consistent across strains is appropriate, but perhaps the results would have been different had the experiments been conducted in a different order. Performing a new set of experiments using a different order could address this possibility; alternatively, it may also be possible to address it in the discussion section.

-Some strains appear to show much more (or less) variability than others on particular behavioral tests. It might be valuable to quantify this variability, as it could indicate how to interpret the results of a given test for a particular strain. For example, it seems notable that in Fig. 1A, C57BL6 mice exhibit much lower variability than, e.g., 129SvEv mice.

-The figure and table legends are missing.

-What is shown on the axes of Fig. 5D?

-It is not clear why the color scheme in Fig. 6A differs from that in the preceding figures.

-Fig. 8 seems out of order and isn't referred to in the Results section. Also, it is not clear whether there is any significance to the markers being colored red in this figure. Does this correspond to the color scheme in other figures?

-There are numerous spelling errors, particularly in the Results section. Just as one example, p. 11 line 3 (significnatly, distnace, accross).

Gidon Felsen

Department of Physiology and Biophysics

University of Colorado School of Medicine

(I am including my name/affiliation because I have opted in to open peer review)

Reviewer: 2

Comments to the Author(s)

Comments here indexed by [page].[line]

3.23 "...few inter-strain comparisons are available." This is not really true in general. Perhaps you need to be more specific? Unfortunately, failure to acknowledge the vastness of the literature (or postponement to the Discussion as here) immediately raises a question in any reader's mind about the value of your work. I am not saying your work is not valuable, but failure to put it in the context of the large available literature deprives your work of much of its potential value in the mind of the reader. In case you find that your work is largely not original, that is not a real problem, since – and I think this is an important and useful aspect of journals like RSOS – “studies testing the reproducibility of significant work are encouraged”. You just need to be open about this and highlight what part of your work is new and what is confirmatory. In any case, the four references provided are only a small subset of the literature available, which addresses the problem of background variation on two levels. First, many papers, stretching back many years, address various aspects of genetic background and its contribution to differences in wildtype behaviour. Second, beyond the issue of wildtype comparisons, there are reports of interactions of induced mutations with genetic background. Many of these are anecdotal (single gene or behaviour), but a couple of recent systematic studies were done, one by Sittig et al (many mutants in multiple backgrounds, *Neuron*, 2016) and one by van de Lagemaat et al (three mutants in C57BL/6J vs 129S5, *Genes Brain and Behaviour*, 2017). There are at least ten other references in the latter paper looking at one or the other aspect of genetic background in mouse behaviour. I also note you have more papers referenced further along in your manuscript.

5.28 boli? Call them fecal boli

5.33 put ref 19 with ref 20.

6.49 is it alterations or alternations? Please be consistent.

7.44 snuggly is not the same as snugly, the first of many typos.

8.23 Kruskal

9.9 in 7.5 lines of writing, 8 typos. Out of respect for your reviewers, proofreading for spelling, punctuation and grammar are essential!

10.33 second half of the sentence is hard to follow and appears to jump to the conclusion that mice produce faecal boli due to anxiety alone. Better to use the phraseology "observation X suggests interpretation Y" rather than "we observed X because Y".

10.37 What is the p-value of the correlation in Fig 8A? By looking, I doubt it is significant. A quick check in R tells me the p-value for your rank correlation is 0.2333. You need more data (one data point per animal?) to prove a correlation. Furthermore it is pretty clear that this is a strain effect. Does this correlation (if it exists) also exist (i.e. achieve significance) within a strain?

10.52 129 mice exhibit less alternations than other strains. What might explain that? Could this be because 129s are inactive?

11.3 129 also show greater perseveration. Why? Might this be because 129s are just inactive (Fig 1)?

11.6 How did you determine difficulty of 129 mice in changing arms? Or might it be they just didn't care, or were too anxious? How do you differentiate between these possibilities?

11.18 129 mice often exhibit bimodal activity, very active or very inactive, and this may change within a day. C57BL/6J are much more active. When you then parameterise the performance of an individual mouse by percentage of visits to the correct arm of the Y maze, your measures of 129 mice will vary widely, and this will decrease your power to detect a C57BL/6J-129 difference. You need to understand and lead the reader through this.

11.38, 11.49 general tendency to inactivity in 129 mice is well known and predicts that they will stop swimming or struggling sooner without needing to invoke any "depression". You should mention this confounding effect and address how significant your results are after accounting quantitatively for this prediction.

12.18 Where did DISC1 (presumably k/o) mice come from? What background? What dataset were you using? Or did you assess a cohort of mice? What methods did you use? What were the controls? What was the hypothesis? Why were DISC1 mice not mentioned before? Where are the relevant statistical methods? How does this fit into the context of the paper?

12.32 it seems surprising that 129 mice should have second-fewest stress calls when literature suggests 129 strains generally are more anxious. Why is that? What confounds might account for it?

13.20 How did you choose your animals for PCA? Choosing animals opens the door to gross bias and forbids any interpretation of these results (unless you can provide an algorithm demonstrating that these choices were made blind). Please remove this section from the paper.

13.43 You appear to jump to the conclusion that because wildtype mice differ in performance in these tasks, mouse models of human disorders (which usually involve some induced mutation) will also be affected. While this is certainly true at the level of gross measures (cm travelled, cm/s speed, etc), the work of van de Lagemaat et al (2017) shows that this may not be the case, even for backgrounds as different as C57BL/6J and 129S5, when one considers the Cohen d effect size of the mutation. In other words, although the backgrounds may exhibit different baselines and scales for behavioural measures, the behavioural change in mutants relative to wildtypes may

nevertheless be in the same direction and with the same number of standard deviations in both strains.

15.10 here you finally mention Disc1. If it is important, move it up to the introduction.

15.26 on what basis do you argue that bimodal behaviour suggests variable penetrance? Is the behaviour mode an immutable characteristic of an individual animal? How did you test this?

15.35 from the previous comment, you go on to say “other factors” than genetic background play a crucial role in developing an animal’s behavioural phenotype. If lower mobility (for example in 129) were actually an immutable characteristic of an individual mouse, you might make this claim. However, you have not tested this.

16.3 Again, because you have chosen a subset of mice for your PCA, the reader should ask if the observed clusters are just an artefact of which animals were chosen. Probably you should remove the PCA from the paper.

16.20 Are the results in this paper a useful baseline for further studies in neurological disorders? People have been saying this for years, but when really carefully tested in light of mutation effect sizes (number of standard deviations between mutant and wildtype) and random sampling, the effect of background strain seems less important. I would not say UN-important, but less so. You should address this in your manuscript too.

Overall conclusions:

Overall, there may be some good data here. It suffers from several major problems and needs a lot of work before it is ready for publication:

- 1) In its initial form, the fact that in the introduction the manuscript largely ignores a huge literature by very big-name individuals over many years is annoying and biases the informed reader against the manuscript in its current form.
- 2) The experienced reviewer is further annoyed by the poor quality of the spelling and grammar (although this is worse in some places than others). Figure 8 is out of order too. These show general sloppiness. To address this, use a good grammar and spell checker and proofread your manuscript (and have others proofread it) before sending it out. Your reputation as a scientist depends in part on this.
- 3) There is a lack of sophistication in your work in that it lacks proper controls/baseline measures (e.g. home cage behaviour).
- 4) There is a general lack of sophistication in your interpretation of the data, leading you to jump to conclusions along the lines of commonly held views. This is also done without citing the relevant papers. To address this, you should, first, read lots of papers in this field and seek to internalise the careful language and arguments of others. It will help you make your own arguments.
- 5) You nowhere mention the sex of the animals. Even if it has no impact on these measures (see your Moy et al reference), it is generally a big issue in mouse models, so you should mention it and mention how you tested/analysed it. If you didn’t test the effect of sex or use only one sex, it greatly undermines your whole experiment.
- 6) Your study performs several assays on wildtype mice from five strains. Unfortunately, your methods leave many questions unanswered. For example, in the sociability test is similar to one done with five strains (a different set of strains) by others with much more finesse (i.e. including olfactory tests and comparing to home cage behaviour) 14 years ago. That study was cited by you (Moy et al, 2004). With time, we expect methods to improve, but this is not the case here. I think you have made a good start, but you need a lot more controls and explanation before this paper is ready for review or publication.

Reviewer: 3

Comments to the Author(s)

In the manuscript the authors aim to assess and draw attention to phenotypic differences in inbred mouse strains. After a series of behavioral assessments conducted over 16 days the authors conclude that there is a large variability in the response of different strains to various behavior tasks and recommending that future experiments including these strains should be aware of these differences prior to and during the interpretation of any results acquired through the use of these mice as disease models.

The results of these studies draw attention to the importance of considering mouse strain when designing your experiments. This concept is touched upon in other publications, but is not always considered by experimenters. These studies are a good reminder of the importance of this and provides some direct comparisons of common mouse strains that can aid experimenters in their selection.

The behavior tasks utilized in this study seem to be carried out well and assessed well.

Experiments also appear to be spread out over a 16 day period and ordered in a manner that places the most stressful experiments towards the end, controlling for the impact of stress external to the desired behavioral experiment impacting the performance of the mice

Minor Criticisms:

-Author should emphasize a little more that any number of differences could be responsible for their behavioral performance, besides just nicotinic binding sites and DISC1

-Authors should bring to attention other published differences found in 129SvEv line, as this is a line they have chosen to emphasize, and how it compares to their results, for example: previous studies regarding fear and extinction

- the use of CD1 as an outbred strain is a good one, but no discussion is made comparing the CD1 to the inbred strains and what that means.

-In the results section the results given by one strain is referred to as "normal" while the behavior may be the anticipated result, the authors should refrain from the use of the word "normal" as behavioral is considered "normal" only based on previous studies that have utilized specific inbred mouse lines themselves (usually B6)

-there are many spelling errors throughout the manuscript that needs to be addressed, particularly with the misspelling of the word "significant" and "significantly" which occurs at several points throughout the manuscript.

Author's Response to Decision Letter for (RSOS-181399.R0)

See Appendix A.

RSOS-190574.R0

Review form: Reviewer 1 (Gidon Felsen)

Is the manuscript scientifically sound in its present form?

Yes

Are the interpretations and conclusions justified by the results?

Yes

Is the language acceptable?

Yes

Is it clear how to access all supporting data?

Yes

Do you have any ethical concerns with this paper?

No

Have you any concerns about statistical analyses in this paper?

No

Recommendation?

Accept with minor revision (please list in comments)

Comments to the Author(s)

The authors have addressed most of my previous comments and improved the manuscript. One remaining minor comment is that Figure 8 still seems to be out of order: Fig. 8A is referred to after Fig. 2, and Fig. 8B is referred to after Fig. 5.

Review form: Reviewer 2

Is the manuscript scientifically sound in its present form?

Yes

Are the interpretations and conclusions justified by the results?

Yes

Is the language acceptable?

Yes

Is it clear how to access all supporting data?

Yes

Do you have any ethical concerns with this paper?

No

Have you any concerns about statistical analyses in this paper?

No

Recommendation?

Accept with minor revision (please list in comments)

Comments to the Author(s)

A much improved, much easier to read manuscript, but a few minor issues still exist.

Comments below refer to page_no.line_no

4.49 criterion

11.3. Fig 1D, significance bar missing for C57BL/6J-BALB/c comparison

11.28 "Fecal boli production was also analyzed (Fig. 1G) and was found to vary among strains, with the greatest number produced by 129SvEv" Should this be CBA?

11.33 I like the correlation analysis of anxiety-related behaviour

12.6 Make sure you reference Fig 2B, C, D in the main text; Correct the legend of Figure 2: include description of incorrect entries and a description of panel D.

12.21 Fig 3, make sure you reference panels A and B in the legend.

13.15 Figure 8 is still out of place. Include the sentence beginning "In addition..." several pages before, and include Figure 8 in Figure 1

13.20 Ref to fig 8B is out of place

14.13 You don't comment on Fig 6b, so drop it from the figure.

14.21 Restructure sentence beginning "There was a...". Suggest "At 30 ms and 75dB there were significant differences between A and B ($P < 0.001$), C and D ($P < 0.05$), and E and F ($P < 0.05$).". Note most significant result first.

14.21 In legend of Fig 7, clarify the x-axis labels, with a statement like "X-axis labels represent duration (30 vs 100 ms) and intensity (75 vs 85 dB)". Also, clarify what the bars represent: mean? Include error bars, or better, plot bee-swarm plots as in other figures, showing all data for individual mice.

14.38 Your PCA is remarkably clustered, but I was surprised that in the n=8 case there are always only 8 mice shown, and in the n=6 case, there are always only 6 mice shown. Did you manually choose which mice you included? If so, the analysis is biased, and I would ask you to please remove it. If you did not choose the mice, please mention this with a statement like, "Remarkably, exactly 6 of 12 mice of each line completed all assays, and these are shown in Fig 9B".

Decision letter (RSOS-190574.R0)

07-May-2019

Dear Dr Sultana

On behalf of the Editor, I am pleased to inform you that your Manuscript RSOS-190574 entitled "Contrasting characteristic behaviors among common laboratory mouse strains" has been accepted for publication in Royal Society Open Science subject to minor revision in accordance with the referee suggestions. Please find the referees' comments at the end of this email.

The reviewers and Subject Editor have recommended publication, but also suggest some minor

revisions to your manuscript. Therefore, I invite you to respond to the comments and revise your manuscript.

- **Ethics statement**

- **Data accessibility**

If you wish to submit your supporting data or code to Dryad (<http://datadryad.org/>), or modify your current submission to dryad, please use the following link:
<http://datadryad.org/submit?journalID=RSOS&manu=RSOS-190574>

- **Competing interests**

- **Authors' contributions**

- **Acknowledgements**

- **Funding statement**

Please note that we cannot publish your manuscript without these end statements included. We

have included a screenshot example of the end statements for reference. If you feel that a given heading is not relevant to your paper, please nevertheless include the heading and explicitly state that it is not relevant to your work.

Because the schedule for publication is very tight, it is a condition of publication that you submit the revised version of your manuscript before 16-May-2019. Please note that the revision deadline will expire at 00.00am on this date. If you do not think you will be able to meet this date please let me know immediately.

Kind regards,
 Andrew Dunn
 Royal Society Open Science Editorial Office
 Royal Society Open Science

on behalf of Dr Ryan Y Wong (Associate Editor) and Kevin Padian (Subject Editor)
openscience@royalsociety.org

Associate Editor Comments to Author (Dr Ryan Y Wong):

Dear Dr. Sultana,

Thank you for your submission and addressing the feedback from the previous reviewers. I am glad to say that I am recommending acceptance pending minor revisions as suggested by the appended reviews.

Reviewer comments to Author:

Reviewer: 2

Comments to the Author(s)

A much improved, much easier to read manuscript, but a few minor issues still exist.

Comments below refer to page_no.line_no

4.49 criterion

11.3. Fig 1D, significance bar missing for C57BL/6J-BALB/c comparison

11.28 "Fecal boli production was also analyzed (Fig. 1G) and was found to vary among strains, with the greatest number produced by 129SvEv" Should this be CBA?

11.33 I like the correlation analysis of anxiety-related behaviour

12.6 Make sure you reference Fig 2B, C, D in the main text; Correct the legend of Figure 2: include description of incorrect entries and a description of panel D.

12.21 Fig 3, make sure you reference panels A and B in the legend.

13.15 Figure 8 is still out of place. Include the sentence beginning "In addition..." several pages before, and include Figure 8 in Figure 1

13.20 Ref to fig 8B is out of place

14.13 You don't comment on Fig 6b, so drop it from the figure.

14.21 Restructure sentence beginning "There was a...". Suggest "At 30 ms and 75dB there were significant differences between A and B ($P < 0.001$), C and D ($P < 0.05$), and E and F ($P < 0.05$).". Note most significant result first.

14.21 In legend of Fig 7, clarify the x-axis labels, with a statement like "X-axis labels represent duration (30 vs 100 ms) and intensity (75 vs 85 dB)". Also, clarify what the bars represent: mean? Include error bars, or better, plot bee-swarm plots as in other figures, showing all data for individual mice.

14.38 Your PCA is remarkably clustered, but I was surprised that in the $n=8$ case there are always

only 8 mice shown, and in the n=6 case, there are always only 6 mice shown. Did you manually choose which mice you included? If so, the analysis is biased, and I would ask you to please remove it. If you did not choose the mice, please mention this with a statement like, "Remarkably, exactly 6 of 12 mice of each line completed all assays, and these are shown in Fig 9B".

Reviewer: 1

Comments to the Author(s)

The authors have addressed most of my previous comments and improved the manuscript. One remaining minor comment is that Figure 8 still seems to be out of order: Fig. 8A is referred to after Fig. 2, and Fig. 8B is referred to after Fig. 5.

Author's Response to Decision Letter for (RSOS-190574.R0)

See Appendix B.

Decision letter (RSOS-190574.R1)

14-May-2019

Dear Dr Sultana,

I am pleased to inform you that your manuscript entitled "Contrasting characteristic behaviors among common laboratory mouse strains" is now accepted for publication in Royal Society Open Science.

Kind regards,
Royal Society Open Science Editorial Office
Royal Society Open Science

on behalf of Dr Ryan Y Wong (Associate Editor) and Kevin Padian (Subject Editor)
openscience@royalsociety.org

Appendix A

Reviewer: 1

Sultana and colleagues compare performance on standard behavioral tasks between several commonly-used mouse strains. They show some significant differences in behavior, demonstrating that caution is warranted when interpreting comparative data collected in multiple strains. The manuscript is fairly straightforward and the quantification of differences in behavior across strains could provide a valuable resource to the community. However, there are several concerns about the manuscript as currently written:

-We thank the reviewer for their comments, which have been used to improve the manuscript. We have used their comments in their entirety and we believe that the manuscript is much improved as a result. We have included a version of the manuscript with revisions tracked.

The authors make it sound as if researchers often attempt to compare results across strains. This would clearly be a problem, but how common is it? Does it occur in primary research studies, and/or in reviews attempting to synthesize the literature? Citing some papers in which this sort of problematic comparison is made, and how that has negatively affected the papers' conclusions, would bolster the rationale for the current manuscript.

-We agree with the reviewer for the useful insight. As suggested, regarding the comparisons in review and primary studies where these comparisons have been made, the references have been added in the introduction of manuscript.

-Is it possible that behavior was affected by the order in which the tests was performed? Keeping the order consistent across strains is appropriate, but perhaps the results would have been different had the experiments been conducted in a different order. Performing a new set of experiments using a different order could address this possibility; alternatively, it may also be possible to address it in the discussion section.

- We agree with the reviewer about the concern for the order of behavioral tests, we closely followed the protocol by <https://www.mousephenotype.org/impress> and other similar studies mentioned in manuscript, where most stressful tests were performed at the end, and to avoid retention/latency problems the tests done on the same apparatus were performed in separate groups of animals (for e.g. Spontaneous alternation and Y maze task for working memory)

-Some strains appear to show much more (or less) variability than others on particular behavioral tests. It might be valuable to quantify this variability, as it could indicate how to

interpret the results of a given test for a particular strain. For example, it seems notable that in Fig. 1A, C57BL6 mice exhibit much lower variability than, e.g., 129SvEv mice.

-We agree with the authors observation. We have addressed this variability in the discussion. We have also used standard error of mean was calculated and depicted in the figures to determine and exhibit intra-strain variability

-The figure and table legends are missing.

- We apologize for any confusion. The figure and table legends alongside the figures were uploaded as separate files in previous submission.

-What is shown on the axes of Fig. 5D?

-We made the corrections as suggested and the axes title have been added.

-It is not clear why the color scheme in Fig. 6A differs from that in the preceding figures.

- As suggested we changed the color scheme of Fig. 6A to match all other figures in the manuscript.

-Fig. 8 seems out of order and isn't referred to in the Results section. Also, it is not clear whether there is any significance to the markers being colored red in this figure. Does this correspond to the color scheme in other figures?

-The figure is mentioned in the results section. There was no significance to the red color of the markers, we have changed these to solid black to avoid any confusion to the readers.

-There are numerous spelling errors, particularly in the Results section. Just as one example, p. 11 line 3 (significnatly, distnace, accross).

-As suggested, a complete and thorough revision was performed in order to (hopefully) rectify all the spelling and grammar errors.

Reviewer: 2

Comments to the Author(s)

Comments here indexed by [page].[line]

3.23 “...few inter-strain comparisons are available.” This is not really true in general. Perhaps you need to be more specific? Unfortunately, failure to acknowledge the vastness of the literature (or postponement to the Discussion as here) immediately raises a question in any reader’s mind about the value of your work. I am not saying your work is not valuable, but failure to put it in the context of the large available literature deprives your work of much of its potential value in the mind of the reader. In case you find that your work is largely not original, that is not a real problem, since – and I think this is an important and useful aspect of journals like RSOS – “studies testing the reproducibility of significant work are encouraged”. You just need to be open about this and highlight what part of your work is new and what is confirmatory. In any case, the four references provided are only a small subset of the literature available, which addresses the problem of background variation on two levels. First, many papers, stretching back many years, address various aspects of genetic background and its contribution to differences in wildtype behaviour. Second, beyond the issue of wildtype comparisons, there are reports of interactions of induced mutations with genetic background. Many of these are anecdotal (single gene or behaviour), but a couple of recent systematic studies were done, one by Sittig et al (many mutants in multiple backgrounds, Neuron, 2016) and one by van de Lagemaat et al (three mutants in C57BL/6J vs 129S5, Genes Brain and Behaviour, 2017). There are at least ten other references in the latter paper looking at one or the other aspect of genetic background in mouse behaviour. I also note you have more papers referenced further along in your manuscript.

--We thank the reviewer for their comments, which have been used to improve the manuscript. We have used their comments in their entirety and we believe that the manuscript is much improved as a result. We have included a version of the manuscript with revisions tracked. As suggested by the reviewers we have tried to address their concerns and highlight the relevant literature as we progress in the manuscript.

5.28 boli? Call them fecal boli

-We have corrected ‘boli’ with ‘fecal boli’

5.33 put ref 19 with ref 20.

-We have put ref 19 with ref 20

6.49 is it alterations or alternations? Please be consistent.

-We have made the changes as suggested and to keep it consistent replaced alterations with alternations

7.44 snugly is not the same as snugly, the first of many typos.

-We have corrected snugly with snugly.

8.23 Kruskal

-We have corrected this to: Kruskal

9.9 in 7.5 lines of writing, 8 typos. Out of respect for your reviewers, proofreading for spelling, punctuation and grammar are essential!

-We apologize for the numerous typos. We have (hopefully) eliminated all of these errors in spelling, punctuation and grammar throughout the manuscript.

10.33 second half of the sentence is hard to follow and appears to jump to the conclusion that mice produce faecal boli due to anxiety alone. Better to use the phraseology “observation X suggests interpretation Y” rather than “we observed X because Y”.

-As suggested, we have reworded this section.

10.37 What is the p-value of the correlation in Fig 8A? By looking, I doubt it is significant. A quick check in R tells me the p-value for your rank correlation is 0.2333. You need more data (one data point per animal?) to prove a correlation. Furthermore, it is pretty clear that this is a strain effect. Does this correlation (if it exists) also exist (i.e. achieve significance) within a strain?

- Fig. 8A, the p value is not significant as noted by the reviewer due to the fewer number of data points, but the r value of Spearman rank correlation shows a moderate correlation, whereas when calculated with all the data sets (i.e. one point per animal for individual strains) none of the strains, except BALB/c shows a positive significant correlation.

-Fig 8B, there exists a positive significant correlation between the individual animals' mobility time in the forced swim and tail suspension tests. We have noted these in the results.

10.52 129 mice exhibit less alternations than other strains. What might explain that? Could this be because 129s are inactive?

- We understand reviewer's concern, but the calculations are based on percent entries to the stated zones, rather than the absolute numbers. Furthermore, each strain acts as its own control in a way that a threshold has been set for all of them i.e. only the animals making a total of or more than 10 entries to all the arms were considered qualified for inclusion.

11.3 129 also show greater perseveration. Why? Might this be because 129s are just inactive (Fig 1)?

- Again, as with the above, we only included animals that were active and made a minimum of ten entries.

11.6 How did you determine difficulty of 129 mice in changing arms? Or might it be they just didn't care, or were too anxious? How do you differentiate between these possibilities?

-We have removed this statement.

11.18 129 mice often exhibit bimodal activity, very active or very inactive, and this may change within a day. C57BL/6J are much more active. When you then parameterize the performance of an individual mouse by percentage of visits to the correct arm of the Y maze, your measures of 129 mice will vary widely, and this will decrease your power to detect a C57BL/6J-129 difference. You need to understand and lead the reader through this.

-Again, we understand the reviewer's concern regarding the activity of the 129S mouse. We only included those animals that exhibited enough exploratory behavior during the closed arm phase in this analysis.

11.38, 11.49 general tendency to inactivity in 129 mice is well known and predicts that they will stop swimming or struggling sooner without needing to invoke any "depression". You should mention this confounding effect and address how significant your results are after accounting quantitatively for this prediction.

-We agree with the reviewer and have revised this section to note this potentially confounding effect here. Our results though corroborate previous experiments, such as those carried out by Gomez-Sintes *et al.* 2014, demonstrating other depressive-like symptoms in this strain.

12.18 Where did DISC1 (presumably k/o) mice come from? What background? What dataset were you using? Or did you assess a cohort of mice? What methods did you use? What were the controls? What was the hypothesis? Why were DISC1 mice not mentioned before? Where are the relevant statistical methods? How does this fit into the context of the paper?

-We apologize for this typographic error. This has been changed to 129SvEv.

12.32 it seems surprising that 129 mice should have second-fewest stress calls when literature suggests 129 strains generally are more anxious. Why is that? What confounds might account for it?

-We agree that this is surprising. We suggest that the stress calls in this case are fewer because they were recorded when the animal was put under stress of tail suspension, also the threshold intensity for filtering the calls was more than 30kHz with a minimum duration of 5ms. The other confound could be due to the time of the day when the recordings were done (in the first half of the day).

13.20 How did you choose your animals for PCA? Choosing animals opens the door to gross bias and forbids any interpretation of these results (unless you can provide an algorithm demonstrating that these choices were made blind). Please remove this section from the paper.

-All animals that were behaviorally tested were added for the PCA. The distinction between the two PCA analyses is that some animals were removed from the Y-maze analysis, due to not meeting the threshold for number of entries. Thus, one analysis includes all animals that completed all tests, while the other includes all animals that completed the tests, except for the Y-maze.

13.43 You appear to jump to the conclusion that because wildtype mice differ in performance in these tasks, mouse models of human disorders (which usually involve some induced mutation) will also be affected. While this is certainly true at the level of gross measures (cm travelled, cm/s speed, etc), the work of van de Lagemaat et al (2017) shows that this may not be the case, even for backgrounds as different as C57BL/6J and 129S5, when one considers the Cohen d effect size of the mutation. In other words, although the backgrounds may exhibit different baselines and scales for behavioral measures, the

behavioral change in mutants relative to wildtypes may nevertheless be in the same direction and with the same number of standard deviations in both strains.

-We completely agree and thank the reviewer for their insightful comment. We have acknowledged and incorporated the suggested concept in the manuscript to give readers another view of dealing with the problem of influence of genetic background on the behavior of animals. We have also added a concluding paragraph that emphasizes this point. This is also true that the consideration of Cohen d effect size of mutation can help us avoid the existing problem, but it might require us to increase our sample size when the mutations on two different backgrounds are compared. Here in our study we are primarily trying to re-enforce the pre-existing notion that there is a behavioral difference among the animals based on their genetic background.

15.10 here you finally mention Disc1. If it is important, move it up to the introduction.

-As suggested, we have introduced the Disc1 mutation in the Introduction.

15.26 on what basis do you argue that bimodal behavior suggests variable penetrance? Is the behavior mode an immutable characteristic of an individual animal? How did you test this?

-We have removed this statement.

15.35 from the previous comment, you go on to say “other factors” than genetic background play a crucial role in developing an animal’s behavioral phenotype. If lower mobility (for example in 129) were actually an immutable characteristic of an individual mouse, you might make this claim. However, you have not tested this.

-We agree that we have not tested the immutability of these behaviors in these strains. We have tempered our statements to remove any speculative statements.

16.3 Again, because you have chosen a subset of mice for your PCA, the reader should ask if the observed clusters are just an artefact of which animals were chosen. Probably you should remove the PCA from the paper.

- We understand the reviewer’s concern, but we included all the animals to perform PCA, as noted above,

16.20 Are the results in this paper a useful baseline for further studies in neurological disorders? People have been saying this for years, but when really carefully tested in light

of mutation effect sizes (number of standard deviations between mutant and wildtype) and random sampling, the effect of background strain seems less important. I would not say UN-important, but less so. You should address this in your manuscript too.

- We completely agree with the reviewer's comments. As per the suggestion by the reviewer we have added a concluding paragraph that addresses this important issue directly.

Overall conclusions:

Overall, there may be some good data here. It suffers from several major problems and needs a lot of work before it is ready for publication:

1) In its initial form, the fact that in the introduction the manuscript largely ignores a huge literature by very big-name individuals over many years is annoying and biases the informed reader against the manuscript in its current form.

-Again, we thank the reviewer for their comments. We apologize for any omissions and have revised the manuscript to include many sources that were previously omitted.

2) The experienced reviewer is further annoyed by the poor quality of the spelling and grammar (although this is worse in some places than others). Figure 8 is out of order too. These show general sloppiness. To address this, use a good grammar and spell checker and proofread your manuscript (and have others proofread it) before sending it out. Your reputation as a scientist depends in part on this.

-Again, we apologize for the poor quality of editing in the prior draft. We have attempted to remove all such errors in this resubmission.

3) There is a lack of sophistication in your work in that it lacks proper controls/baseline measures (e.g. home cage behaviour).

-We thank the reviewer for their comments and have addressed this in the resubmission.

4) There is a general lack of sophistication in your interpretation of the data, leading you to jump to conclusions along the lines of commonly held views. This is also done without citing the relevant papers. To address this, you should, first, read lots of papers in this field and seek to internalise the careful language and arguments of others. It will help you make your own arguments.

-We agree with the reviewer and have revised the discussion in the revision.

5) You nowhere mention the sex of the animals. Even if it has no impact on these measures (see your Moy et al reference), it is generally a big issue in mouse models, so you should

mention it and mention how you tested/analysed it. If you didn't test the effect of sex or use only one sex, it greatly undermines your whole experiment.

-We apologize for this omission and have included the genders used in this resubmission. We used mice of both genders and analyzed for gender effects, but did not observe any differences on these tests. We have noted this in the manuscript.

6) Your study performs several assays on wildtype mice from five strains. Unfortunately, your methods leave many questions unanswered. For example, in the sociability test is similar to one done with five strains (a different set of strains) by others with much more finesse (i.e. including olfactory tests and comparing to home cage behaviour) 14 years ago. That study was cited by you (Moy et al, 2004). With time, we expect methods to improve, but this is not the case here. I think you have made a good start, but you need a lot more controls and explanation before this paper is ready for review or publication.

Again, we thank the reviewer and have attempted to address their criticisms in the revision.

Reviewer: 3

Comments to the Author(s)

In the manuscript the authors aim to assess and draw attention to phenotypic differences in in-bred mouse strains. After a series of behavioral assessments conducted over 16 days the authors conclude that there is a large variability in the response of different strains to various behavior tasks and recommending that future experiments including these strains should be aware of these differences prior to and during the interpretation of any results acquired through the use of these mice as disease models.

The results of these studies draw attention to the importance of considering mouse strain when designing your experiments. This concept is touched upon in other publications but is not always considered by experimenters. These studies are a good reminder of the importance of this and provides some direct comparisons of common mouse strains that can aid experimenters in their selection.

The behavior tasks utilized in this study seem to be carried out well and assessed well. Experiments also appear to be spread out over a 16 day period and ordered in a manner that places the most stressful experiments towards the end, controlling for the impact of stress external to the desired behavioral experiment impacting the performance of the mice

-We thank the reviewer for their comments, which have been used to improve the manuscript. We have used their comments in their entirety and we believe that the manuscript is much improved as a result. We have included a version of the manuscript with revisions tracked.

Minor Criticisms:

-Author should emphasize a little more that any number of differences could be responsible for their behavioral performance, besides just nicotinic binding sites and DISC1

-As suggested, we have tried to cite the literature making the potential reasons for differences more lucid.

-Authors should bring to attention other published differences found in 129SvEv line, as this is a line they have chosen to emphasize, and how it compares to their results, for example: previous studies regarding fear and extinction

-As suggested by the reviewer, relevant literature and text is added.

- the use of CD1 as an outbred strain is a good one, but no discussion is made comparing the CD1 to the inbred strains and what that means.

-Discussion regarding the behavior difference of CD1 mouse is added.

-In the results section the results given by one strain is referred to as “normal” while the behavior may be the anticipated result, the authors should refrain from the use of the word “normal” as behavioral is considered “normal” only based on previous studies that have utilized specific inbred mouse lines themselves (usually B6)

-The changes are made to the text, and caution was observed while using comparative statements.

-there are many spelling errors throughout the manuscript that needs to be addressed, particularly with the misspelling of the word “significant” and “significantly” which occurs at several points throughout the manuscript.

-As suggested, the corrections in spelling errors are made.

Appendix B

Reviewer: 2

A much improved, much easier to read manuscript, but a few minor issues still exist.

-We thank the reviewer for their comments, which have been used to improve the manuscript. We have used their comments in their entirety and we believe that the manuscript is much improved as a result.

Comments below refer to page_no.line_no

4.49 criterion

-We made the corrections as suggested.

11.3. Fig 1D, significance bar missing for C57BL/6J-BALB/c comparison

-We made the corrections as suggested.

11.28 “Fecal boli production was also analyzed (Fig. 1G) and was found to vary among strains, with the greatest number produced by 129SvEv” Should this be CBA?

-As suggested, the correction has been made.

11.33 I like the correlation analysis of anxiety-related behaviour

-We thank the reviewer for the appreciation.

12.6 Make sure you reference Fig 2B, C, D in the main text; Correct the legend of Figure 2: include description of incorrect entries and a description of panel D.

- We thank the reviewer for taking note, the correction has been made.

12.21 Fig 3, make sure you reference panels A and B in the legend.

-As suggested, both the panels (A and B) for Fig. 3 have been referenced.

13.15 Figure 8 is still out of place. Include the sentence beginning “In addition...” several pages before, and include Figure 8 in Figure 1

-As suggested, we have corrected the placement and discussion of this figure.

13.20 Ref to fig 8B is out of place

-As suggested, we have corrected the placement and discussion of this figure.

14.13 You don’t comment on Fig 6b, so drop it from the figure.

- As indicated, we have made a mention about Fig. 6B in the main text.

14.21 Restructure sentence beginning “There was a...”. Suggest “At 30 ms and 75dB there were significant differences between A and B ($P < 0.001$), C and D ($P < 0.05$), and E and F ($P < 0.05$).” Note most significant result first.

-As suggested, we have reworded this section.

14.21 In legend of Fig 7, clarify the x-axis labels, with a statement like “X-axis labels represent duration (30 vs 100 ms) and intensity (75 vs 85 dB)”. Also, clarify what the bars represent: mean? Include error bars, or better, plot bee-swarm plots as in other figures, showing all data for individual mice.

-We thank you for your kind suggestion, axis title has been added to the figure caption. Figure depicts the mean, and for more clarity to the readers error bars have been added.

14.38 Your PCA is remarkably clustered, but I was surprised that in the $n=8$ case there are always only 8 mice shown, and in the $n=6$ case, there are always only 6 mice shown. Did you manually choose which mice you included? If so, the analysis is biased, and I would ask you to please remove it. If you did not choose the mice, please mention this with a statement like, “Remarkably, exactly 6 of 12 mice of each line completed all assays, and these are shown in Fig 9B”.

-As suggested we have added the above sentence in the main text, and Fig. 6A ($n=8$) has been removed to avoid any bias.

Reviewer: 1

Comments to the Author(s)

The authors have addressed most of my previous comments and improved the manuscript. One remaining minor comment is that Figure 8 still seems to be out of order: Fig. 8A is referred to after Fig. 2, and Fig. 8B is referred to after Fig. 5.

-As suggested, we have corrected the placement and discussion of these figures.